# Using Data Tools and Systems to Drive Change in Early Childhood Education for Disadvantaged Children in South Africa

**DOI:** 10.3390/children10091470

**Published:** 2023-08-28

**Authors:** Sonja Giese, Andrew Dawes, Linda Biersteker, Elizabeth Girdwood, Junita Henry

**Affiliations:** 1DataDrive2030, Westlake, Cape Town 7945, South Africa; linda.biersteker2@gmail.com (L.B.); elizabeth@datadrive2030.co.za (E.G.); junita.h13@gmail.com (J.H.); 2Psychology Department, University of Cape Town, Rondebosch, Cape Town 7700, South Africa; andrew.dawes@uct.ac.za

**Keywords:** South Africa, early childhood education, ECE, LMIC, Early Learning Outcomes Measure, ELOM, data systems

## Abstract

In line with United Nations Sustainable Development Goal (SDG) 4.2, South Africa’s National Development Plan commits to providing high-quality early childhood education to all children by 2030 to drive improved child outcomes. Prior to 2016, South Africa lacked reliable, locally standardised, valid, and cross-culturally fair assessment tools for measuring preschool quality and child outcomes, suitable for use at scale within a resource-constrained context. In this paper we detail the development and evolution of a suite of early learning measurement (ELOM) tools designed to address this measurement gap. The development process included reviews of literature and other relevant assessment tools; a review of local curriculum standards and expected child outcomes; extensive consultation with government officials, child development experts, and early learning practitioners, iterative user testing; and assessment of linguistic, cultural, functional, and metric equivalence across all 11 official South African languages. To support use of the ELOM tools at scale, and by users with varying levels of research expertise, administration is digitised and embedded within an end-to-end data value chain. ELOM data collected since 2016 quantify the striking socio-economic gradient in early childhood development in South Africa, demonstrate the relationship between physical stunting, socio-emotional functioning and learning outcomes, and provide evidence of the positive impact of high-quality early learning programmes on preschool child outcomes. To promote secondary analyses, data from multiple studies are regularly collated into a shared dataset, which is made open access via an online data portal. We describe the services and support that make up the ELOM data value chain, noting several key challenges and enablers of data-driven change within this context. These include deep technical expertise within a multidisciplinary and collaborative team, patient and flexible capital from mission-aligned investors, a fit-for-purpose institutional home, the appropriate use of technology, a user-centred approach to development and testing, sensitivity to children’s diverse linguistic and socio-economic circumstances, careful consideration of requirements for scale, appropriate training and support for a non-professional assessor base, and a commitment to ongoing learning and continuous enhancement. Practical examples are provided of ways in which the ELOM tools and data are used for programme monitoring and enhancement purposes, to evaluate the relative effectiveness of early learning interventions, to motivate for greater budget and inform more effective resource allocation, to support the development of enabling Government systems, and to track progress towards the attainment of national and global development goals. We share lessons learnt during the development of the tools and discuss the factors that have driven their uptake in South Africa.

## 1. Introduction

This paper describes the development of the Early Learning Outcome Measure (ELOM) suite of assessment tools. These tools are designed to address the need for locally standardised and robust instruments which produce fair assessments of preschool-aged children from across the ethnolinguistic and socio-economic spectrum and can be affordably used at scale within the South African context. They assess a range of early developmental outcomes in young children, as well as measuring the quality of the learning environment in home and early learning programme (ELP) settings. The tools are digitised and embedded within an end-to-end data value chain that enables users with varying levels of research expertise to be able to collect and use the data to drive change.

The ELOM tools and systems are firmly located within the broader early childhood education ecosystem in South Africa. Practical examples are provided of ways in which the tools are used for programme monitoring and enhancement purposes, for research, and for tracking progress towards the attainment of national and global development goals. To promote secondary analyses, data from multiple studies are regularly collated into a meta dataset which is made open access via an online data portal. We describe the services and support that make up the ELOM data value chain, noting some of the challenges and enablers of data-driven change within a resource constrained context.

The motivating force behind the development of the ELOM tools and systems is the desire to close the opportunity gaps in early childhood outcomes in South Africa, where significant inequities are evident in the early learning opportunities and outcomes of children from different socio-economic backgrounds. Poor children (the majority in the country) are less likely to have access to an ECE programme than their better-off peers of the same age [1], and when they do, these programmes tend to be of lower quality than those serving children from higher-income households [2]. Poor children are more likely to start school without the necessary learning foundations in place [3], and most attend under-resourced primary schools that are unable to offer the quality of education required to help them catch up. This sets most of our children up for failure, with long term implications for them and for the country as a whole.

The data that are generated using the ELOM suite of tools enable us to agitate for greater and smarter investment in services to enhance access to quality ECE programmes for all children. Our goals are ultimately to increase the percentage of young children who are on track for their age when it comes to physical, socio-emotional, and cognitive development, and to decrease the performance gap between young children in the wealthier and poorer households at the point of entry into school.

## 2. Overview of South Africa’s ECE System

South Africa defines early childhood development (ECD) as “an umbrella term that applies to the policies by which children from birth to at least nine years grow and thrive, physically, mentally, emotionally, spiritually, morally and socially” [4] (paragraph 73). This paper focuses on the early childhood education (ECE) component of ECD. Services for the 0–4 years age group are provided primarily by non-profit organisations and micro-social enterprises.

Prior to South African Democracy in 1994, access to early education opportunities was segregated by ethnicity and was generally limited—especially for black children, the majority of whom lived in under-resourced rural areas [5]. In 1992, 7% of black children aged 0–6 years were estimated to be enrolled in ECE centres compared to 33% of white children [6].

The first post-Apartheid democratic government recognised ECD as a key area in the process of reconstruction and development. The 1995 Education White Paper 1 [1 above], identified early childhood as the starting point for human resource development. The Paper committed the state to providing 10 years of free and compulsory schooling, starting with a reception (kindergarten) year for five-year-olds. After a period in which ECD policy and models for the delivery of the reception year were piloted, Education White Paper No. 5 [7] provided for the largest ever South African public sector policy commitment to ECD in the universal roll-out of a national system of provision for the reception year, known as Grade R, for children aged five years. For children aged 0–4 years, White Paper 5 prioritised the development of a strategic plan for inter-sectoral collaboration. This was developed into the National Integrated Plan for ECD Zero to Four Years, which was published in 2005 [8], overseen by the Interdepartmental Committee for ECD, and coordinated by the Department of Social Development since 2008. In 2012, ECD was elevated to a priority in the National Development Plan Vision 2030 [9].

A Diagnostic Review of ECD [10] which mapped existing services for health, welfare, and education, and identified service gaps, was undertaken as a first step to policy development for 0–4-year-olds. In 2015, the National Integrated ECD Policy [11] was accepted by Cabinet. The Policy goal is to have integrated “quality ECD programmes for all infants and young children and their caregivers” by 2030 [p. 49]. The National Curriculum Framework for Children Zero to Four Years [12], also finalised in 2015, is informed by nationally derived and validated early learning and development standards [13]. In April 2022, responsibility for coordination of ECD moved from the Department of Social Development (DSD) to the Department of Basic Education (DBE), and the focus on better early learning outcomes has consequently increased.

### 2.1. Increasing Access to ECE Programmes

The reception year (Grade R) is included in the Foundation Phase of the Schooling Curriculum and Assessment Policy Statement [14], but is not currently compulsory; although, this will change once new legislation, the Basic Education Laws Amendment Bill, is passed by parliament. Grade R enrolment is free for children from poor backgrounds and has allowed for a large influx of these children into the education system, notably in the largely rural Eastern Cape, Free State, and Limpopo provinces [15]. Enrolment in Grade R in ordinary public (state funded) and independent schools has increased over the years. By the end of 2012, 75% of Grade 1 children enrolled in public schools had attended Grade R [16]. While enrolment gains were negatively impacted by the COVID-19 Pandemic [17], in 2021, it was estimated that 80% of five year olds were enrolled in Grade R or in an ECE programme [15].

Analysis of General Household Survey (GHS) 2018 data [18] show that programme participation by the preschool age group is skewed toward older children. Only 57% of children aged three years were enrolled in an ECE programme compared to 70% of 4 year olds. Participation is also skewed by income. A three-year-old child in the richest quintile is almost twice as likely to be enrolled in an ECE programme as their same-age peers in the poorest quintile [1], as shown in Figure 1.

One of the reasons for lower enrolment rates amongst poorer and younger children is the fact that there is no public ECE provision for 0–4 year olds. These services are offered mostly by community-based non-profit organisations and microenterprises and almost all providers charge fees, with a reported national average monthly fee of ZAR 509 per month [19] (USD 1.00 = ZAR 19.34). Non-profit ECE service providers are partially funded via a means-tested, per-child government subsidy of ZAR 17.00 (<USD 1.00) per child, per day. Programmes are eligible to receive these funds when they comply with specific health, safety, and registration requirements in accordance with the Children’s Act 38 of 2005 as amended. Many service providers are unable to comply with the requirements for registration and are therefore unable to access the subsidy [20]. Analyses by the South African ECD research organisation, Ilifa Labantwana, show that unregistered ECE programmes, mostly operating in poor communities, provide early learning to about 2.5 million children [21], and in 2018, the ECE programme attendance of only 626,000 children (less than 10% of children aged 0–5 years) was subsidised by the state. Even when a programme is subsidised, caregivers are typically required to pay fees to cover the operational cost shortfalls [19].

### 2.2. South African ECE Data

The 2015 National Integrated ECD Policy mandates the collection of child, service, and impact data, and the development of a national research agenda to measure the impact of ECE programmes on child outcomes and national development goals. However, access to reliable and current data to inform evidence-based planning and service improvement has been identified as a challenge since 2001, when efforts to scale up ECE programmes and quality began in earnest [10,22,23]. This includes data on service supply and quality, enrolment data, and data on service impact (child outcomes).

The Education Management Information System (EMIS) unit in the DBE has included information on Grade R enrolment in public and independent schools since 2005, but not on Grade R in non-school-based sites such as community-based programmes. For children aged 0–6 years, national surveys such as the National Income Dynamics Study [24] and the General Household Surveys include some questions on ECD participation and enable analysis of trends in ECD participation over different ages and across time.

A Nationwide Audit of ECD Provisioning was conducted in 2000 [25], with a further audit of centre-based provision in 2013 [26]. In preparation for the ECD function shift from DSD to DBE in April 2022, the DBE commissioned a National Census of Early Learning Programmes (ELPs). This sought to identify and document every ECE programme (registered and unregistered) across the country, a significant effort to remedy understanding of provisioning to inform ECE planning [19].

Data on the impact of ECE programmes are particularly limited. However, Annual National Assessments (ANAs) of Language and Mathematics have indicated poor outcomes and a marked social gradient in performance by Grade 3, which increases over subsequent grades. Participation in grade R is related to improved achievement in the ANAs in later grades, but is negligible for children in the three poorest school quintiles, due to poor teaching and learning quality and systems failure [27].

In 2021, a nationally representative sample of early learning programmes was visited to collect child-level outcome data on the proportion of 4–5-year-old children on track in three key areas of development: early learning, social-emotional functioning, and physical growth. This inaugural Thrive by Five Index is the largest national level assessment of child outcomes undertaken in South Africa [3]. The Index was enabled by the availability of the standardised population measures developed in 2016—the Early Learning Outcomes Measure (ELOM) suite of tools.

## 3. The Need for Tools, Data, and Systems

South Africa ratified the Sustainable Development Goals (SDGs) in 2016. SDG 4.2. requires a commitment to report on the proportion of children under 5 years of age who are developmentally on track in health, learning, and psychosocial well-being. However, no nationally standardised tools for the measurement of SDG Target 4.2 were available.

In 2003, Daniel Wagner [28] urged the development of alternative approaches to assessment in Low- and Middle-Income Countries (LMICs) that would be smaller, quicker, and cheaper. In his 2011 volume [29], he captures our intent in designing the ELOM tools: “What if researchers began with a focus on the actual learning needs of disadvantaged children in poor schools, and designed assessments from that vantage point?” (p. 15). He argued for smaller, quicker, cheaper (SQC) assessments able to pinpoint the nature of children’s difficulties and to do so with a precision and timing that will allow for potential intervention before it is too late (p. 141). While his focus was on literacy, Wagner’s points are relevant to the development of assessment tools for LMICs that cover all early developmental domains needed to inform policy and intervention at scale. For him, “*smaller*” assessment methods may include the use of smaller populations “just large enough” to generate robust data, such as children attending particular programmes or children in a local area. Smaller also applies to the assessment design. One conclusion from Wagner’s position is that instruments need not attempt to cover developmental domains comprehensively and in depth, but rather have sufficient items to provide reliable and robust assessments that are appropriate for answering key questions pertinent to programme improvement as well as to national and local policy levels. He urges that data should be produced “*quicker*”, so that action can be taken to affect programmes and policy implementation in as short a time as possible. In our South African experience, building relationships with key stakeholders at programme and government level has been key to gaining acceptance for our approaches to both individual child and programme quality assessment, and also to obtaining buy-in for large-scale surveys designed to generate data to inform policy for preschool children. Finally, we have heeded Wagner’s call for “*cheaper*” assessment methods. His 2011 contribution was inspired by efforts “to promote the use of quality learning indicators in education, in the framework of Education for All and the UN Millennium Development Goals” (p. 20), so that educational outcomes in developing countries might be improved. Finally, he notes (p. 15), that “assessments that are centered on the needs and requirements of industrialised and well-resourced countries might be less than suitable for use in developing country contexts where the learning situation varies in important and discernible ways”. This crucial observation has been repeated in the cross-cultural developmental literature and by African developmental scholars over many years, e.g., [30,31].

Prior to 2016 and the development of the Early Learning Outcomes Measures (ELOM), South Africa did not have locally standardised instruments in all the official languages suitable for assessing the development of preschool-age children that could be administered economically and at scale [32]. No developmental tests were available for which linguistic, cultural, functional, and metric equivalence was established for children from all South Africa’s ethnolinguistic and socio-economic backgrounds.

Mwaura and Marfo [33] note Africa’s historical “dependence on imported instruments, often adopted with little or no adaptations” (p. 138). With few recent exceptions [34,35], tests developed in the majority world have been used in African research [36]. Rather than being standardised in local languages, they require translation in the field, which increases the probability of measurement error.

Tests commonly used for individual assessment of young children in South Africa are restricted to licenced psychologists (a significant cost driver) and are not standardised in the local languages. Examples include the Wechsler Preschool and Primary Scale of Intelligence (WPPSI-IV) [37] and the Griffiths Mental Development Scales (both kits and administration requirements are costly). While the McCarthy Scales [38] and the Griffiths [39] have been adapted for use among some South African communities and a local developmental testing is available [40], translation and standardisation have not been undertaken and administration is lengthy.

In an effort to address the measurement gap in South Africa, the development of the ELOM suite of tools began in 2016.

## 4. The Development of a Suite of Early Learning Measurement Tools

The suite of ELOM tools developed thus far is displayed in Figure 2.

As will be evident, five of the ELOM tools are for child assessment, one assesses programme quality, and one assesses the home learning environment. These are described briefly below. Their uses are covered in a later section.

### 4.1. Child Assessment Tools

Our inspiration in designing the ELOM child assessment tools was very similar to that of Wagner and sub-Saharan African colleagues: to produce affordable and robust instruments aligned to South Africa’s National Curriculum Framework for children from birth to 4 years, the Grade R (kindergarten) Curriculum Assessment Policy Standards (CAPS) (the ELOM 4&5 Years Assessment Tool), and the Grade 1 mathematics and language CAPS (the ELOM 6&7 Literacy and Mathematics Assessment Tools). A further objective was to develop tools that could be administered by trained non-professionals in under an hour, and which produced fair assessments of children from across the ethnolinguistic and socio-economic spectrum [41]. In the first instance, the goal was to design reliable tools that ECE programmes and the education authorities could use to assess whether children were developmentally on track for Grade R and for Grade 1. In a country with eleven official languages, this would prove to be a challenging task. In addition, test design would need to consider the fact that 63% of children live in income poverty [42] in a country with the most unequal income distribution globally (Gini Coefficient = 63.0) [43]. Language, cultural, and economic variation would mean differential exposure of children to the sorts of materials and tasks commonly used in developmental assessments.

The ELOM 4&5 Years Assessment Tool is designed to assess whether children in the age group 50–69 months are ready to learn in Grade R. The tool measures functioning in five domains: Gross Motor Development (GMD), Fine Motor Coordination and Visual Motor Integration (FMC-VMI), Emergent Numeracy and Mathematics (ENM), Cognitive and Executive Functioning (CEF), and Emergent Literacy and Language (ELL). Most of the 23 items in this tool were drawn from existing instruments and are aligned to the developmental outcomes specified in the National Curriculum Framework for children from birth–4 years. The tool development process included the standardisation and establishment of functional and metric equivalence across the home languages of 70% of South Africa’s children. It is essential to ensure that a test designed for the assessment of children from several ethnolinguistic backgrounds provides a fair measure of their ability. Questions of bias and equivalence are fundamental issues that must be addressed [44,45].

New measures must be assessed for their linguistic, cultural, functional, and metric equivalence. For comparative purposes across cultures, the underlying constructs being measured must be understood in the same way and demonstrate the same psychological (factor) structure, and scale items must enable all children, regardless of their background, to demonstrate their true ability or attitude. Bias is evident when the items are formulated in a manner that the instrument makes children from particular ethnolinguistic backgrounds more or less likely to demonstrate their true competencies. These points were addressed in the development of both the ELOM 4&5 and 6&7 Prior to the piloting stage, items were carefully assessed by local experts in child development. Once selected, they were reviewed by preschool and kindergarten teachers working in diverse cultural communities to ascertain whether the test items and materials would require modification. Items were then double translated and adapted following International Test Commission Guidelines and piloted. Following final adaption, psychometric analysis was conducted. Particularly important was analyses of Differential Item Functioning (DIF) to establish whether item difficulty varied across language groups. Using these procedures, cross-language equivalence [41] has been established (children of the same ability score the same regardless of their language) as well as construct, age, and concurrent validity (with the WPPSI-IV), and test–retest reliability [46]. The ELOM 4&5 is now available in all 11 official languages and can be administered by trained assessors who are not registered health professionals or psychologists. Based on their total scores for the test as a whole and in each domain, children are classified into one of three bands: achieving the expected developmental standard (On Track), Falling Behind the standard, and Falling Far Behind the standard (originally known as ‘At Risk’). The expected developmental standard was finalised in consultation with the Department of Basic Education and other national government and academic stakeholders.

Social and emotional functioning is assessed using brief teacher rating scales [47] rather than through observations by independent assessors. Teachers who have known the child for some time are able to provide more reliable ratings (even though subjectivity and bias can be an issue). The ELOM Social-Emotional Rating Scale has two sections, each with six items drawn from existing instruments and factor analysis has shown them to be independent reliable subscales. A full account of the development and properties of the ELOM 4&5 Assessment Tool and the ELOM Social-Emotional Rating Scale is available [47].

The ELOM 4&5 Years Targeting Tool [48] is a short form of the full instrument, comprising one item from each of the five developmental domains. These items were selected through logistic regression with Receiver Operating Characteristic (ROC) [49,50] sensitivity and specificity analyses that showed they distinguished between children who were Falling Far Behind the expected ELOM standard and those that were not. This instrument is used by community-based ECE programmes to identify children in particular need of early learning support.

The ELOM 6&7 Years Assessment Tool is designed to assess early grade literacy and mathematics abilities. Underlying cognitive skills associated with competencies in these areas are included. Item construction was informed by South Africa’s Grade R and Grade 1 CAPS and research into the Grade 1 entry skills that are predictive of reading and mathematics abilities during the early primary phases. Two separate but complementary instruments have been developed: the ELOM 6&7 Years Mathematics Assessment Tool (18 items) and the ELOM 6&7 Years Literacy Assessment Tool (10 items). Both incorporate cognitive skills items. Some items were sourced from existing tests while others were purpose designed by experts in the two areas. These tools are administered by trained assessors and have assessment times of 20–45 min each. The ELOM 6&7 Years Mathematics Assessment Tool covers areas predictive of mathematics abilities in the early years of primary school, such as number sense operations and relationships, sorting and grouping, shape and space, and patterning. The ELOM 6&7 Years Literacy Assessment Tool covers areas predictive of literacy and language abilities in the early years of primary school (including vocabulary, phonics, comprehension, letter recognition and writing skills). Construct validity and reliability have been established for each of mathematics and language. Rasch analysis to investigate differential item functioning (DIF) and item difficulty in each of South Africa’s official languages is currently being finalised.

Development of both the ELOM 4&5 Years Assessment Tool and the 6&7 Years Assessment Tools was dependent on forming strong relationships with early childhood specialists and with government officials responsible for the foundation phase of schooling. Furthermore, a key requirement was the donor funding needed for all phases of the development process as government funding was not available.

Preliminary steps in the development process included reviews of literature, including developmental tests used in the sub-Saharan region, e.g., [51] and international efforts to establish measures appropriate for use at scale in low- and middle-income countries, for example the International Development and Early Learning Assessment (IDELA) [52], the East Asia-Pacific Early Child Development Scales (EAP-ECDS) [53] and the Measuring Early Learning Quality & Outcomes (MELQO) initiative [54]. Consultations were held with local and international experts and, particularly importantly, with early education practitioners and officials. A review of curriculum standards and expected outcomes for key areas of child development was also undertaken. Thereafter, tools were piloted to determine if there were any aspects of administration that needed adjustment including assessor difficulties in administration, children not understanding instructions, whether the test kit and use of tablet-based administration was appropriate in all contexts, and the reliability of the translations into the various languages.

Validating these tools across a range of language and class backgrounds and across a range of service delivery contexts is a challenging but essential step. The demographics of the target population need to be carefully considered. In any multilingual country (and there are few exceptions today), it is essential that close attention is paid to translation. We drew on the International Test Commission (ITC) guidelines [55] in this regard, using double translations undertaken by local language experts, educators, and community informants rather than professional translation services, as these were often likely to produce translations that were too “high level” and out of touch with local common language usage. In several languages, we were confronted with a range of local usages which rendered one translation from the English inadequate. This was particularly the case for expressive language and vocabulary items. In the latter instance, more than one correct answer from a short, standardised list of alternatives is acceptable (additions to the list is an ongoing process).

When new measures are to be deployed at scale for assessing children from a variety of ethnolinguistic backgrounds, it is essential that they be subjected to assessment for systematic measurement error and bias. It is important to ensure as little bias as possible in the design, administration, and scoring of the instrument by establishing measurement equivalence. Children speaking different languages, and children living in remote rural villages whose socialisation and developmental affordances are very different to those who have resided in modern urban environments since birth or early childhood, must be given equal opportunity to demonstrate their abilities. The challenge is not just translation, although that is not a simple matter in itself, but adaptation [56]. For example, the materials used to test a skill may be more familiar to the urban group than the rural child; the difficulty of items may vary across groups—for example, a word in a productive vocabulary test may be easier in the English version than the isiZulu, so the order of word presentation has to differ for the two languages. And of course, the entire testing situation may be more familiar to some young children than others [44,57]. These key points were considered when developing the ELOM tools.

### 4.2. Learning Programme Quality Tool

Knowing children’s skills at the commencement of Grade R and Grade 1 is essential, but if we are to design improvements likely to lead to better outcomes, we also need data on the quality of their early learning programmes. That was the motivation for the development of the Learning Programme Quality Tool [58,59], which is used in group ECE programmes for children aged 3–6 years. The tool is aligned with South African programme guidelines and assesses commonly measured indicators of the learning setting and interactions associated with good child outcomes. Factor analysis has established five subscales: Learning Environment, Assessment for Learning and Teaching, Relationships and Interactions, Curriculum, and Teaching Strategies.

### 4.3. Home Learning Environment Tool

The relationship between children’s home environment and children’s cognitive development and school achievement is well established. Family socio-economic status and caregiver education are particularly powerful predictors of these outcomes, as is the quality of stimulation provided to preschool children by caregivers and other household members during early childhood [60].

There are several well-established tools for assessing the dimensions of children’s home environments. Examples include the Home Observation for Measurement of the Environment (HOME) [61] and the PICCOLO (Parenting Interactions with Children: Checklist of Observations Linked to Outcomes) [62]. Where home observations are not feasible due to cost or limited observer expertise, as well as in large-scale surveys, short interview methods are an alternative.

The ELOM Home Learning Environment Tool is a short questionnaire that is designed for administration to the child’s main caregiver (parents or other household members primarily responsible for the child). It captures key features of a young child’s home learning environment that are known to be associated with early language and numeracy abilities, as well as cognitive functioning. As an instrument appropriate for brief caregiver interviews was needed, the Home Learning Environment Tool drew on two well-recognised sources: an instrument developed by Melhuish and colleagues in the United Kingdom [63], and the Early Childhood Development module of the UNICEF Multiple Indicator Cluster Survey For Children Under Five (MICS) [64]. These tools include a limited number of items used in the HOME.

Factor analysis of the ELOM Home Learning Environment Tool has confirmed two factors: resources (e.g., availability of books and play equipment) and activities (activities with the child such as play and telling stories). Caregivers are also asked how much time they have during the week and the weekend for activities with the child [65].

## 5. Systems to Support the Use of ELOM Tools at Scale

Underpinning all our work is the desire to enable habits of data collection and use within the early childhood ecosystem at all levels. Developing tools that are reliable, fair, and contextually appropriate was a necessary first step. However, we recognised early on that to facilitate data-driven change in practice, we would need to build an assessor workforce and embed these tools within a supportive system, enabling use at scale while simultaneously ensuring quality standards.

Through ongoing user testing and engagement and the appropriate use of technology, we have gradually put in place the necessary enablers, effectively supporting users with varying levels of data and research expertise. This process has generated valuable insights into the overall “data value chain” necessary to encourage and facilitate data-driven quality enhancement in early learning in South Africa. Within this data value chain, our tools and systems are designed to support the full data lifecycle from collection to analysis and usage, transforming raw data into actionable insights. By way of example, ELOM tools are digitised, backend systems are codified, real time monitoring is enabled through an online field work management portal, and cleaning and reporting processes are automated wherever possible. All the ELOM tools have accompanying training guides and, in the case of the direct child assessments, can only be administered by accredited assessors. Accreditation is carefully managed to help maintain quality standards. Figure 3 describes the systems that have been built to support the scale of the ELOM tools.

Common challenges faced by data collectors and data consumers are systematically documented. In response, we have produced a series of accessible resources as part of our support toolbox. Examples of products include a guide for field work managers on common field work challenges, code books and benchmark guides, a guide to understanding effect size, a guide to understanding maturation effects, and a brief on approaches to sampling, among others. More recently we have also begun to explore ways of leveraging insights and best practices from behavioural science so as to embed behaviour change principles and practices within the ELOM data value chain, to encourage the use of data to drive change. As an example, all ELOM users are required to sign a user agreement. We are adding a simple action checklist to this agreement, encouraging users to apply their minds to how they plan to use the data once collected. Most users also request an automated report at the end of data collection, summarising key findings and benchmarking their results against national standards and comparative reference groups. We are experimenting with different ways of communicating and visualising findings to test which approaches are most likely to lead to changes in behaviour. We also connect users via communities of practice, to gather feedback to inform ELOM systems enhancements, and to share lessons, insights, and resources that have been shown to be effective in driving improved outcomes.

## 6. Ways in Which the ELOM Tools and Data Are Used

The ELOM tools, and the evidence generated, are now used by a wide range of stakeholders—from community-based ECE service providers to training organisations, researchers, funders, and policy makers. The ELOM tools are also used for a range of purposes, well beyond the scope of what we had originally envisaged when we began this journey. Several examples are provided below. More information on the application of the ELOM tools can be found on the DataDrive2030 website.

### 6.1. Strengthening ELP Design and Quality

Organisations working directly with children collect and use outcomes data to identify and build on the strengths of their programme, and to address areas of weakness. In the South African context, in-house programme monitoring can provide limited data of dubious quality and evaluation is prohibitively expensive when external consultants are engaged. The ELOM tools are designed to make programme monitoring and outcome evaluation more accessible and affordable to organisations who are either providing ECE services or supporting them. We offer customised support to organisations to help them use the tools appropriately and for a variety of purposes. We also assist with data quality assurance, data cleaning, analysis and interpretation, storage, and sharing. Simple standard reporting templates allow organisations to compare their outcomes with peers and against national benchmarks. By automating much of these processes, we are able to standardise reports and keep costs low. Figure 4 and Figure 5 illustrate the type of information included in automated reports.

Along with the graphs, users are provided with brief descriptions of key findings. As an example, Figure 4 shows that only one third of children in this sample are reaching the expected standard for the ELOM Total score, and worryingly, 52% are Falling Far Behind. Children in this programme seem to be making limited progress across most domains and, in particular, more attention needs to be paid to the development of children’s skills in the areas of: Gross Motor development (only 31% of children are on track), Fine Motor Coordination and Visual Motor Integration (only 21% on track), Early Numeracy and Maths (only 19% of children are on track), and Cognition and Executive Functioning (only 27% on track).

**User reports also include domain specific information and comparisons. As an example, Figure 5** displays the average FMC-VMI score for the children in this sample (black star) compared to the national average score (Thrive by Five Index) for the same age children (grey square), and to the average score achieved by children attending programmes charging similar monthly fees (grey diamond). The ELOM ‘on track’ cut-off score for children of this age is illustrated by the green dot. 

The majority of users have drawn on their results to adjust and improve their services—an area where great value has been added. And while these user reports are intended for individual programmes, we are also able to identify trends across programmes, and we share these insights with our communities of practice. As an example, we consistently see poorer outcomes for boys than girls in most domains, and we have observed patterns in performance of rural versus urban children across different developmental domains. We are also able to identify programmes that are achieving extraordinary results within key areas of development, such as cognition and executive functioning, and provide the opportunity for them to share their expertise with others.

### 6.2. Investigating the Effects of ECE Programmes, Home Environments, and Community Influences on Learning Outcomes

The ELOM tools are used to research the relative effectiveness of interventions, which enables broader data-driven decision making and informs resource allocation. As an example, The Early Learning Programme Outcomes Study was the first South African study to examine the relative effectiveness of different programmes that aim to improve the early learning outcomes of young children from low-income backgrounds. The study involved a quasi-experimental baseline (pre-programme) and end-line (post-programme) design, comparing two ECE centre models, a mobile programme and playgroup models. The report identified the relative effectiveness of each intervention, and the factors associated with positive outcomes [66]. A sub study investigated the effects of different levels of playgroup programme exposure (a very affordable option for reaching poor children), showing the value added by attending three in contrast to one or two sessions per week [67]. A significant outcome of this study is that several of the programmes have adapted their models on the basis of these findings.

### 6.3. Individual Neurocognitive Assessment

The ELOM 4&5 Assessment Tool was never intended for use in clinical practice. However, because it is the only South African standardised instrument for which cross-cultural fairness has been demonstrated, and which measures developmental domains influenced by cognitive functions, the tool has drawn the interest of clinicians requiring an individual neurocognitive screening test. Information on the ELOM 4&5 Assessment Tool has recently been included in a compendium of test batteries “for which Africa-based normative data are available” (p. 23). There are few such instruments. The volume brings together data from studies conducted in Africa which have sought to address the challenges inherent in the cross-cultural application of psychological tests [68].

### 6.4. Population-Level Surveys of Child Development Status and ECE Programme Quality

The ELOM tools are used in population-level surveys to monitor and report progress towards the attainment of local and global development goals.

The Thrive by Five Index is the largest survey of preschool child outcomes ever undertaken in South Africa, and is a multi-sector partnership, led by the national DBE and coordinated by DataDrive2030 [3]. Launched in April 2022, the Index is the first (baseline) in a series of surveys that will monitor trends over time in the proportion of children who are on track for their age in three key areas of development: early learning, physical growth, and socio-emotional functioning. Between September and November 2021, the index assessed over 5100 children between the ages of 50 and 59 months, enrolled in 1247 ECE programmes across all nine provinces. Every child was individually assessed in their mother tongue using the ELOM 4&5 Assessment Tool. Practitioners were interviewed about children’s emotional readiness for school and their social relations with peers and adults using the ELOM Social-Emotional Rating Scale, and the children’s heights were measured using portable stadiometers. A baseline assessment was undertaken alongside the Thrive by Five Index. Principal and practitioner interviews were conducted in 545 of the 1247 ECE programmes that participated in the Index. Trained assessors also observed and rated the quality of the learning environment and practitioner–child interactions in each of these sites, using the ELOM Learning Programme Quality Tool. These programme assessments offer additional insights into the type and quality of provisioning in differently resourced areas.

The Index found that only 46% of children attending ECE programmes in South Africa are on track for learning overall, and able to perform the tasks expected of a child their age, 26% are falling behind, and 28% falling far behind. Within the five ELOM 4&5 domains, there was especially poor performance in the FMC-VMI, CEF, and ENM domains (Figure 6). Across all domains except gross motor, girls outperform boys, with 9% more girls on track for learning overall. In summation, more than 50% of children entering Grade R classrooms in South Africa are likely to need some kind of intervention in order to be able to cope. This finding has significant implications for the way in which Grade R is structured as a bridging year.

The Thrive by Five Index reinforces the need for a holistic approach to ECE. The combination of risk factors faced by young children in South Africa places some at a considerable disadvantage. When it comes to early learning, being moderately stunted at age 4 years (Height for Age Z > 2 SD below the World Health Organisation median), is roughly equivalent to being 5 to 6 months behind children with a normal height for age, all other things being equal. These delays may be further compounded by social and emotional issues. The Social-Emotional Rating Scale demonstrated a large effect on learning outcomes—children who met the standard for social-emotional functioning performed better on the ELOM 4&5 Assessment Tool.

The Index was successfully launched in April 2022, with a dedicated website, extensive media coverage, and subsequent targeted engagements with various key stakeholder groups. The Index is frequently referenced by the Minister of Basic Education, has been used to inform provincial and national planning, and to motivate for additional national funding for early learning. The intention is to repeat Index data collection every three years, with four data points by 2030. The data are available for secondary analysis via an open access portal (see below for more information on this).

### 6.5. Quantifying the Relationship between Income and Learning Poverty

An enduring challenge in the South African context is inequality. While there have been many attempts to quantify these inequalities in school and university education and health systems, the Index data presents the first opportunity to quantify early child outcome inequalities by socioeconomic status at national and provincial levels. Using ECE programme fees as a proxy for socio-economic status (SES), we found that 83% of children in high-fee ECE programmes (charging more than ZAR 1751 (USD 91.00) per month) are on track (indicated by the green line in Figure 7), compared to only a third (34%) of learners in programmes charging the lowest fees (less than ZAR 110 (USD 6.00) per month).

The spectrum of outcomes in wealth that is seen in young children in South African pre-schoolers is evidence of vastly different early life experiences across income groups. However, the data also highlight considerable variation in performance between individuals within the same income group. Adopting an asset-based approach, ELOM data are being used to identify characteristics of children, homes, and ECE programmes that are associated with “positive deviance”, i.e., children from poorer households who significantly outperform their peers. This information will be used to inform the design of potentially scalable interventions targeting low fee ECE programmes.

## 7. Building a Shared Dataset for Secondary Analyses and Innovative Applications

Data collected through various ELOM studies are fed into large and growing shared datasets. These geolocated datasets enable us to merge in other big data and to tackle a broad range of questions on the enablers and inhibitors of healthy development in young children. We analyse these data in-house and share datasets through the Datafirst open access repository at the University of Cape Town to generate insights that may be of interest to early learning programme operators, researchers, funders, and government. The datasets have been viewed and downloaded many times, and secondary analyses are generating valuable additional insights. As an illustration, we recently held a machine learning competition to identify which features of an early learning programme predict better learning outcomes (total ELOM score) for children. In addition to useful, practical insights, the competition increased ELOM data exposure to non-traditional data scientists, with participation from 67 countries (including 29 African countries) and had a higher-than-average female participation rate of 19%. ELOM data are also being integrated into training and research curricula in various academic institutions.

## 8. Data as an Enabler of Systemic Reform

While South Africa has a holistic and integrated early childhood development policy, responsibility for the coordination of ECD falls to one government department (initially DSD, with a function shift to DBE in April 2022). However, improving child outcomes cannot be achieved by one department alone, not least because the actual provision of ECE services remains primarily with the non-governmental sector. Furthermore, ELOM data highlight the interconnectedness of areas of development in young children, e.g., height for age and socio-emotional functioning are strongly correlated with child learning outcomes. This demands a collaborative approach, with the necessary interventions, in health care and social services for example, extending well beyond the remit of education and earlier in the child’s development through the first 1000 days.

Within this context, a significant contribution of the ELOM tools and data is to pivot the sector from a siloed, input-based model of provisioning to one focused more clearly on the achievement of child outcomes. This approach has the potential to bring key, and often disparate, stakeholder groups together around the achievement of a common goal which is unifying and focused.

The establishment of DataDrive2030 in 2022 coincided with the launch of the Thrive by Five Index and the national ECD census, as well as the transfer of responsibility for ECE leadership in South Africa, from DSD to DBE. In this respect, the ecosystem is primed for transformative change. New leadership opens opportunities for systems reform, and ELOM data have been used to advocate for greater public (and private) investment in ECD, to test alternative financing models (including the design of a new results based financing framework), and to identify key determinants of quality to be included in a national DBE quality assurance and support system (QASS) for ECD. Lessons from the scale up of the ELOM tools are also being applied to plans for scaling the QASS assessment tools, as part of DBE’s routine ECE quality monitoring and support.

## 9. Key Insights on the Factors Driving Uptake of the ELOM Tools and Data

### 9.1. Having a Dedicated Home for the Tools and the Necessary Resources

The ELOM tools were originally conceptualised and commissioned by Innovation Edge (IE) [69], an impact-first investor focused on solving early childhood challenges in South Africa. IE provided the initial seed capital and facilitated access to other early stage investors and strategic partners (between 2014 and 2021, development costs totaled ZAR 10.5 million/USD 550,000). With growing demand for the tools and a clear need for dedicated focus, the decision was taken in 2020 to transition the ELOM suite of tools into a purpose-built entity with the necessary capacity and skills, strong leadership, a compelling vision and access to social and financial capital. In 2022, DataDrive2030 was established to further develop and scale the use of the tools in a sustainable and impactful manner. The organisation’s theory of change (Figure 8) speaks to the full data value chain described above. Within both IE and DataDrive2030, the core teams are small and partnerships with other institutions within the data ecosystem have been critical to ELOM’s development and success. This includes technical partners with expertise in child development and psychometrics, operational partners with deep knowledge of local contexts, as well as strategic and political partners.

### 9.2. Perceived Legitimacy of the Tools

Throughout their development, quality has been a key consideration in the design of the ELOM tools described in this paper. To that end, reviews of relevant literature were undertaken in order to select potential items with particular attention paid to instruments designed for use at scale in low and middle income countries. Consultations with local and international experts and education officials were held, and the psychometric procedures required to establish reliability, validity, and appropriateness for the range of ethnolinguistic and socio-economic backgrounds were conducted. The processes have resulted in a suite of rigorously designed tools. ELOM tools are standardised for the South African population and have been subjected to review by experts in the field and in the case of the ELOM 4&5, published in peer reviewed journals (as already cited).

Transparency has been an additional consideration. Relevant data are available via open access and all procedures undertaken to finalise each tool are available in manuals posted in the public domain on the DataDrive2030 website. Routine in-house monitoring of data quality is enabled through task differentiation, data visualisation, processing protocols, and spot checks. These processes have occasionally surfaced errors in our coding, cleaning or analysis. Our approach to this has been one of open disclosure, proactively notifying all relevant stakeholders, documenting the error and drafting of procedural guides for limiting recurrence.

### 9.3. Building Enabling Systems to Support the Use of Tools and Data

While data tools are necessary for data-driven change, our experience highlights the importance of having accompanying support systems for the full data value chain, from data collection to analysis, interpretation, and use. With the transition to DataDrive2030, significant investment was made in building these systems, and in the development of a supportive team and accompanying resource materials. The data value chain approach has been critical to ELOM’s successful adoption and scale, and includes digitised backend platforms, built-in quality assurance measures, standardised assessor training and accreditation, in-field user support and automated data cleaning, analysis, and reporting.

### 9.4. Ensuring Contextual Fit

As noted earlier, many of the items within the ELOM tools are not original. They have been drawn and adapted from other well-established instruments. However, this adaptation is key and included alignment with local curricula and policy goals, careful consideration of the lived experiences of South African children across varied contexts, the realities of administration in challenging environments and often without connectivity, the need to match assessor requirements with available skills, and to administer the tools across multiple local languages and dialects.

### 9.5. Demand for Data

The development of the ELOM suite of tools coincided with increasing local and international demand for data, including the need to report against the South African National Development Plan and the Sustainable Development Goal ECD targets. Alongside this, was the impetus from ECD sector donors for grantees to assess whether programmes were effective and were addressing the learning needs of poor children. With the introduction of the ELOM tools, we have also seen greater awareness among ECE programme developers of how child outcome data can inform programme design and implementation and attract further investment. Demand for data is further strengthened through communication strategies that serve to demystify concepts and appeal to audiences with varying levels of technical expertise.

### 9.6. Optimising Shared Data Access and Use

Open access ELOM data from major studies is available on the University of Cape Town’s Datafirst website (https://www.datafirst.uct.ac.za (accessed on 27 March 2023)), providing an additional layer of quality assurance as well as opportunities for secondary analysis. Enabling access to and use of the metadata requires upfront consideration of potential users and possible data applications. Key lessons include the development of a data dictionary, and clarity and consistency in variable names and values, e.g., female should always be coded the same, and all child level variables should have the same prefix (child_*). Constructing easy-to-understand codebooks is essential, with enough contextual information provided (i.e., sources) while still allowing for anonymity. Making all data collection forms available to public users is also helpful, with links between variables and forms. With a crowdsourced growing dataset, it is also important to be clear on varied sampling approaches amongst contributors and transparent about the coding and construction of sampling weights, if relevant. From the outset, contributors need to adhere to clear data collection protocols so as to reduce cleaning and merging challenges down the line. It is also critical to ensure consistency in the application of exclusion criteria, and to have full transparency in instances where criteria are not consistently applied, e.g., where exceptions are made and assessments are permitted outside of the prescribed age range or language. All of this takes time and, wherever possible, we have sought to automate routine processes. As we develop and refine our systems, maintaining open channels for user engagement and feedback is key.

### 9.7. Partnerships

As noted, the ELOM tools were originally developed in partnership with a number of technical experts and other stakeholders from the NGO community, academic institutions, investors, and government. While a consultative development process of this nature has inherent challenges (for example, additional time and expense), there have been significant benefits to this approach, including better quality and more contextually appropriate outputs, and broader buy-in and promotion for use by influential institutions. We have also partnered with organisations across the country to support the training of accredited ELOM assessors (300 plus to date). Almost all are employed by these organisations. Organisations using the ELOM tools contribute to building our database of assessed children, which can then be used for secondary analyses, further psychometric analysis, and for tool improvement. An example is the ongoing collection of data on small area variations in language use to inform refinements in the measurement of productive vocabulary, because some African languageshave multiple words for a particular object or action. We are building a database of commonly used alternatives to the word scored correct in the test instructions. Through inclusion of high frequency alternative correct words, the goal is to ensure greater fairness by not discriminating against children who use commonly accepted alternative words. We place high value on these working partnerships and actively seek feedback from users of the tools and the data, through routine feedback loops, ad hoc surveys, and through our communities of practice, in order to ensure that we remain relevant and continue to add value. To ensure maximum uptake and impact, strategic partnerships are key for flagship projects, such as the Thrive by Five Index, a public private partnership with excellent buy-in from the national Department of Basic Education, private sector, funders and ECE programme operators.

### 9.8. Expectation Management

An important part of facilitating *ethical*, data-driven quality enhancement is the management of expectations to ensure reasonableness and to avoid unintended negative consequences. As an example, certain of the ELOM tools have been used as outcomes measures for outcomes based financing mechanisms. Financing contingent on results has merit as a mechanism for incentivising improvements in programme quality and thereby improving child early learning outcomes. However, there are several key ethical considerations to bear in mind (and manage) when considering such schemes. As such, ongoing and open two-way communication with funders has been key to ensuring the appropriate use of the ELOM suite of tools and data.

## 10. Notable Operational Challenges

To revert back to Daniel Wagner’s call for assessments that would be smaller, quicker, and cheaper, it is worth highlighting some of the challenges faced in designing and supporting reliable administration of such instruments.

As detailed by Wagner, ‘smaller’ can be understood in a variety of ways. To start, many NGOs run programmes with large numbers of children, but it is often the case that relatively few (20 to 40) fall in the age group for which the ELOM has been designed. Monitoring and evaluating programmes with these small numbers is a challenge and the expectations of the organisations need to be carefully managed—in particular, clarity is needed on the sample size necessary for these exercises to be useful to them. Furthermore, South Africa’s ECE provision, in many areas, is characterised by micro-level enterprises of multi-aged children, often confined by space constraints. Thus, frequently, there are simply too few children in one age band within individual centres in target communities for the ELOM Assessment tool to produce useful data.

Small can also refer to smaller organisations or local programmes looking for child outcome data, as opposed to larger research studies or programmes with national reach. Experience has shown that these organisations often have less capacity to manage what can be complex fieldwork. Child outcome data collection requires trained and accredited ELOM assessors, equipped with a standardised kit and a functional data collection device, to travel safely to the child, and conduct reliable assessments on site. There needs to be a match with the child’s home language and that of their assessor, and sufficient children of the correct age range present to satisfy the sample design. While large research studies can employ specialised fieldwork managers to support this process, it more often falls to stretched NGO heads or their monitoring and evaluation officers (rare posts in the ECD NGO space). It is no easy task to engage and contract accredited assessors, manage their daily travel schedule (and accommodation, if necessary), and ensure there are sufficient children, kits, tablets, and data or reliable WIFI available. Each assessment environment is different, and while assessors are trained to minimise the distractions on each child, very often, space is limited, the weather unfavourable, and ECE programme staff unsupportive. When managed by ‘smaller’ organisations, this process is easily derailed. Once organisations have reports on the assessment of their children, they may have limited capacity to understand and act appropriately on their findings. Furthermore, given the understandable pressures from donors to deliver programmes that ‘work’, there is a risk that reports are used to placate them rather than drive real change in programme design and delivery.

‘Quicker’ can similarly refer to many areas of tool use, a good number of which are addressed by the creation of the purpose-built entity, DataDrive2030: quick application time, project setup, reporting post-data-collection. However, collecting data on child outcomes is not that quick. An ELOM 4&5 Years assessment takes, on average, 45 min per child, meaning that an assessor can realistically assess 4 children in a morning session, and approximately 20 children over the course of a working week’s mornings. Practically, this means that there is a trade-off between data collection time and the number of assessors required. While fewer assessors mean fewer logistical challenges and less risk of measurement error, the costs involved in keeping them in the field for longer often outweigh these benefits.

And then finally, ‘cheaper’. This too comes with its challenges. Efforts to enhance the quality and consistency of administration meant that in 2020, the decision was made todigitise administration manuals for each tool and embed all instructions, including stop rules, spoken protocols, and materials required, on SurveyCTO, a digital data collection application. This technology has obvious cost implications for those wanting to conduct assessments. However, the benefits outweigh the costs. The reliability of assessments is significantly enhanced and measurement error is significantly reduced by using these standardised procedures (for each language of administration).

Training accredited assessors for direct assessment on the ELOM 4&5 and 6&7 and supporting their continued skill in administration must be costed. Although assessors do not have to be registered professionals (e.g., psychometrists), the training and accreditation process is rigorous to ensure that the assessments are reliable and that the data generated provide an accurate assessment of the children. Training in the child assessment tools requires four to five full days in-person with at least four opportunities for a supervised child assessment for each trainee. Accreditation is not automatic. In order to be accredited, for a period of 2 years, after which, there is refresher training, assessors need to demonstrate their ability to form a warm relationship with children they assess and to administer the ELOM accurately following all protocols and procedures. At the end of their training, assessors watch a standard video of a child being tested on the ELOM 4&5 or 6&7 by a trainer. They then score the child’s performance. Their score is compared with that of the trainer and to be accredited, there must be 85−90% concordance with the trainer’s score.

Once accredited, continued practice is necessary to maintain administration skills. However, some assessors do not have the opportunity to assess children frequently enough to maintain their skill level. As in-person refresher courses for accredited assessors are expensive, given the vast geographic spread of the assessors, online e-courses have been designed for assessors who have had limited exposure to testing children over a specific period. Unless they sign up for the online course, their accreditation is cancelled.

A section on challenges cannot conclude without a spotlight on language standardisation challenges in a multilingual country. The tools are available in 11 official South African languages and the final, 12th language (South African Sign Language) is planned for 2024. As described above, as in any multilingual country, the necessary attention has been paid to translation and adaptation: local language experts, educators, assessors, and community informants have investigated each spoken instruction by an assessor to the child to ensure accuracy in meaning, spelling and child-appropriate grammar. Despite this, challenges constantly emerge. As an example, with the recent development of the fieldwork management portal, additional quality checks were designed to ensure a match between each child’s home language, the tool administration language, and the assessor’s reported language fluencies. Routine data monitoring has revealed a trend in which assessors were choosing to read the English version of the form and perform their own ad hoc translation. While they were fluent speakers, they were avoiding reading in vernacular language because of the complexities therein. This is a significant risk to protocol standardisation, and every effort is made to address ongoing challenges like this through regular communication with our community of assessors.

## 11. Conclusions

Our paper has documented the journey travelled in developing a suite of early learning assessment tools designed for use in a multi-ethnic LMIC with the highest Gini Coefficient in the world. We have described the factors that have facilitated and inhibited the implementation of these tools for multiple purposes within the South African context.

Beginning in 2016, key facilitating factors included deep technical expertise and visionary leadership within the initial core team, most of whom remain actively involved seven years later. The costs of developing valid, reliable, and contextually appropriate assessment tools are considerable and this initiative would not have been possible without patient and flexible support from mission-aligned investors who were comfortable with the risks involved in supporting an early-stage, untested venture. The collaborative nature of this journey has also been key to its success, with multiple strategic, technical, and operating partners on-boarded as needed during various stages. Having diverse, and at times conflicting, voices in the room has also been enabling. Many key decisions, such as where to set the benchmark for performance band cut-offs, are not black and white, and having input from a number of perspectives ensures that the decision-making process is ultimately defensible. One of the core values of our team is that of continuous learning, and this has been foundational to the ELOM story. We learn from our data, our partners, our assessors and trainers, our experiences of administering the tools across myriad contexts and, most importantly, from our mistakes. The ongoing development and continuous enhancement of our tools and systems remain a priority, even once the tools are “finalised”, e.g., through additional psychometrics and iterations to data collection and cleaning processes made possible by technological advancements.

Technology has been a considerable enabler, allowing us to automate much of what was originally paper-based and manual, and to reduce operating costs. With continuous enhancements in AI and machine learning, there are exciting opportunities to further improve efficiencies in our processes to ensure data quality and accuracy, and to utilise our datasets in innovative ways. Our approach to technological integration has been very pragmatic, using and adapting existing platforms with the necessary interoperability rather than building from scratch, until such time as this level of investment is justified.

Take up of the tools has been amplified by endorsements of influential stakeholders, by open dialogue with critics and by transparency at all stages of our processes. We have also been very intentional about identifying and catering to the specific support needs of different users, including those with advanced research skills and those with very limited experience in data collection and use. Designing, from the outset, for these differentiated user groups has been key to growing our user base. Similarly, thinking in advance about how data will be used is critical to the design of the whole data value chain.

The initial ELOM tools were developed as a project within an organisation (Innovation Edge) with a mandate to support early stage ventures. IE provided the initial home for the tools but was not well positioned to realise their full potential. The decision by the IE board to transition the tools into a fit-for-purpose entity was a key milestone in the ELOM journey, and a springboard to enabling impact at scale. The smooth transition between Innovation Edge and DataDrive2030 was enabled by the continuity in leadership (the IE director established and now leads DataDrive2030), ensuring that institutional memory and key stakeholder relationships were retained.

And finally, from the start, the team involved in the conceptualisation and development of the ELOM suite of tools shared a common vision: to use data to help close the opportunity gaps in early childhood in South Africa, providing all children with opportunities to learn and thrive. This common vision has helped us stay on course during the seven-year journey we have travelled thus far. It has been a filter for decision making at critical junctures, and has served to inspire others to support and join our journey.

## Figures and Tables

**Figure 1 children-10-01470-f001:**
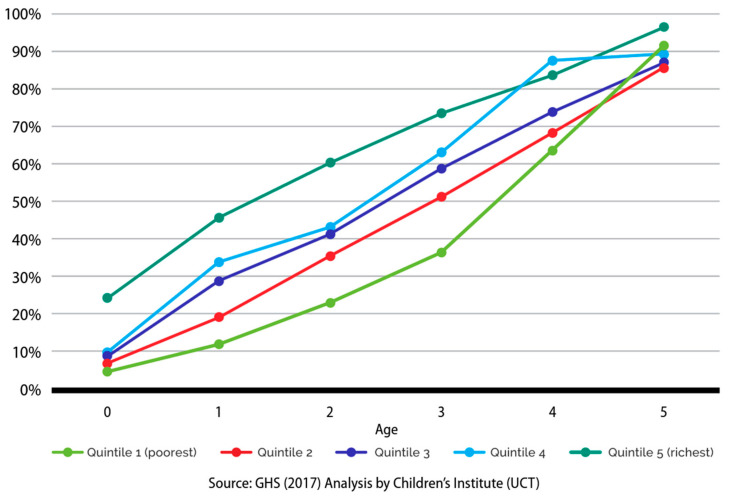
Enrolment in ECE programmes by income and age.

**Figure 2 children-10-01470-f002:**
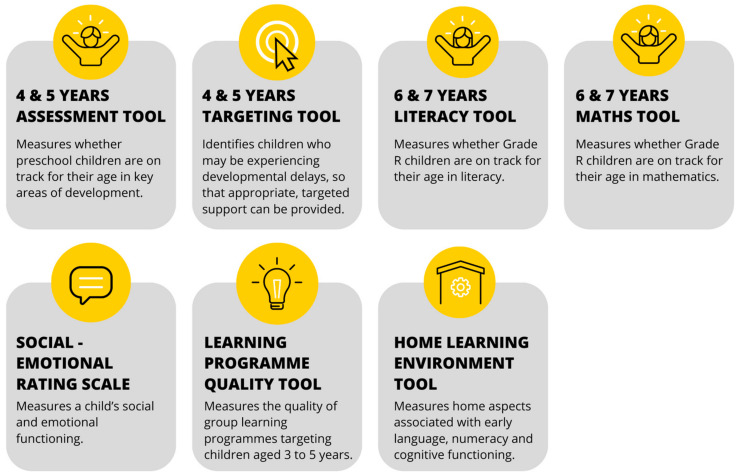
Suite of ELOM tools.

**Figure 3 children-10-01470-f003:**
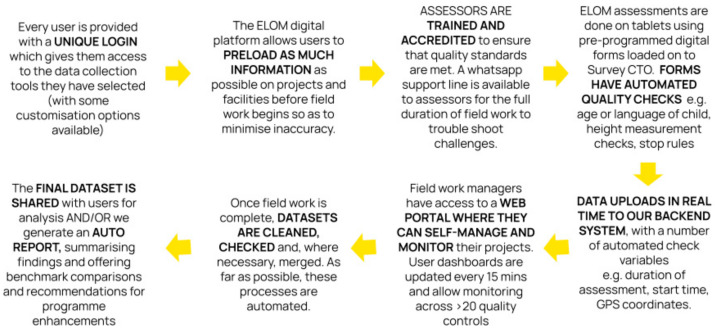
ELOM Tools Support System.

**Figure 4 children-10-01470-f004:**
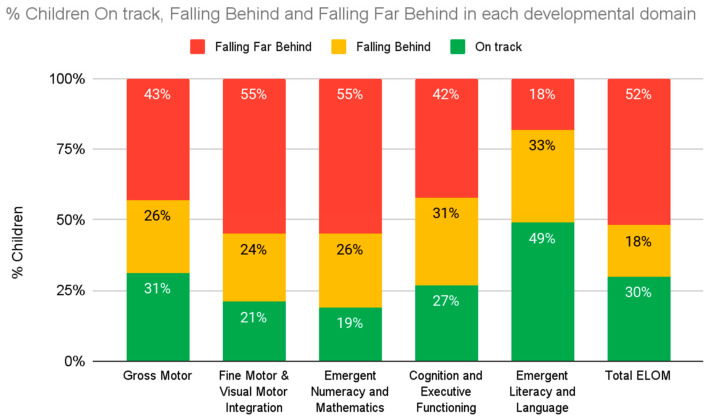
**Example of user report image**: Proportion of children in Programme On Track, Falling Behind or Falling Far Behind in each developmental domain.

**Figure 5 children-10-01470-f005:**
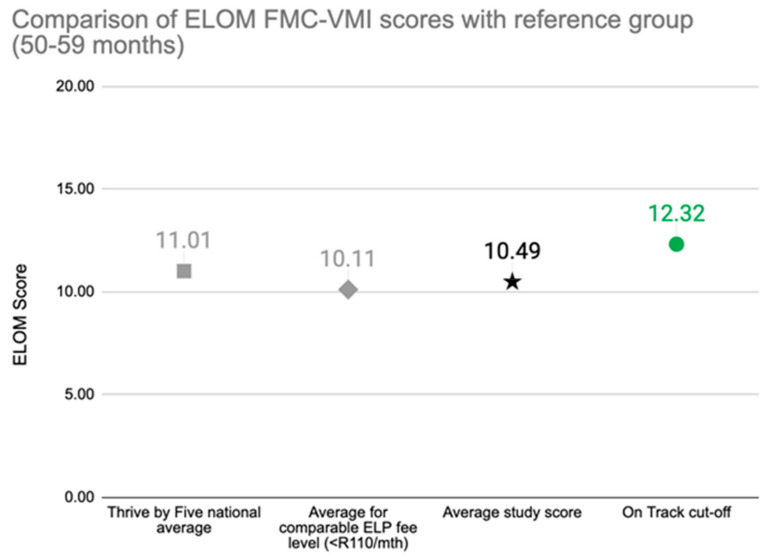
**Example of user report image**: Fine Motor Coordination and Visual Motor Integration Scores of Children in Programme, compared to the National Average.

**Figure 6 children-10-01470-f006:**
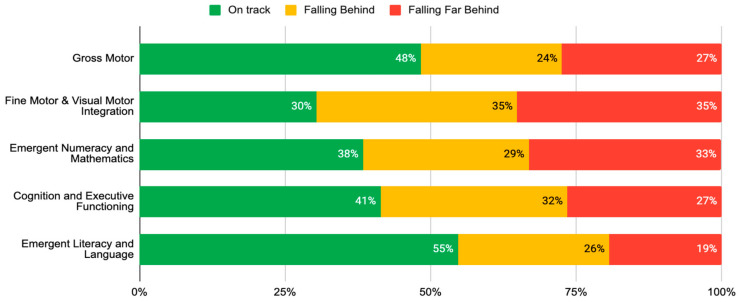
Percentage of children within each performance band for ELOM 4&5 domains.

**Figure 7 children-10-01470-f007:**
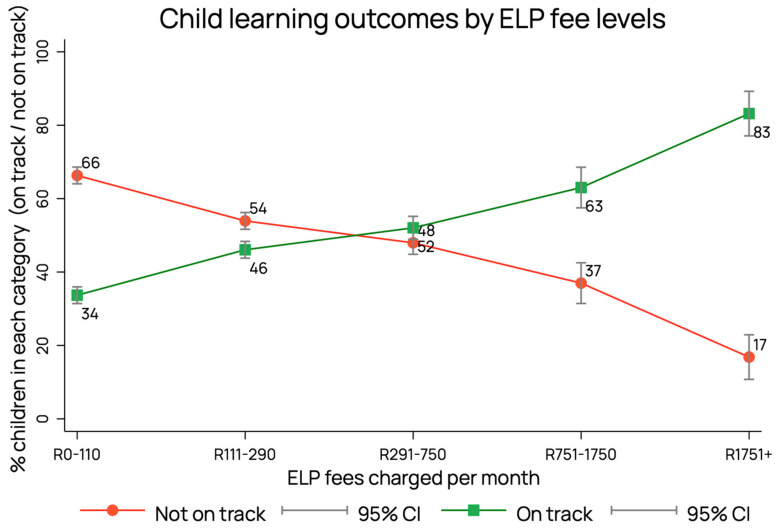
The socio-economic gradient of early learning outcomes in South Africa.

**Figure 8 children-10-01470-f008:**
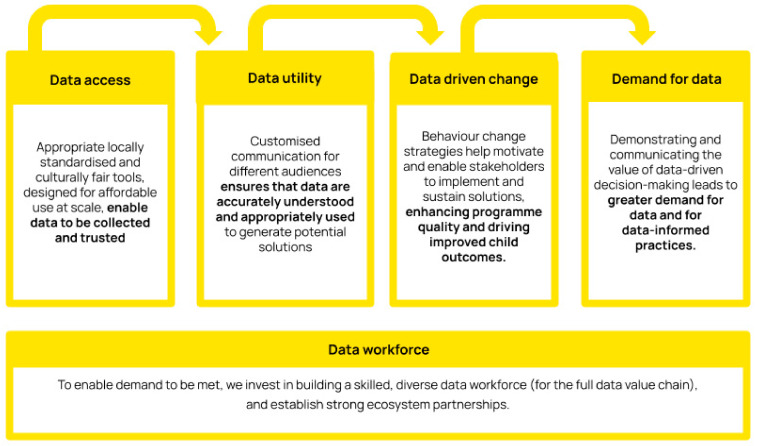
The DataDrive2030 Theory of Change.

## Data Availability

As noted in this paper (lines 554–l559), all data are available at the Datafirst open access repository at the University of Cape Town (https://www.datafirst.uct.ac.za/dataportal/index.php/collections/DD (accessed on 27 March 2023)).

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
