# Peer review of "Using Data Tools and Systems to Drive Change in Early Childhood Education for Disadvantaged Children in South Africa"

_children, 2023, doi:10.3390/children10091470_

Round 1
Reviewer 1 Report
This study details how to build a system that can be used on a large scale, using a set of early learning assessment tools in South Africa as an example. The system utilizes digitized tools to measure a range of developmental outcomes for young children. It provides tools to indicate the quality of the early learning environment at home and in the curriculum set, affirming the unique academic value of this article.
The author mentions five tools for child assessment in ELOM, which are essential to constructing ELOM. The author has also presented the background of these tools in the content. However, the author should go further to introduce the structure of these tools more clearly, which would help the readers to understand the composition of ELOM.
It might be meaningful to see the authors share more projects within the ELOM framework, which is also the most critical part. Through introducing these projects, readers can have a clearer understanding of the areas covered by ELOM and its sub-fields of concern, and hopefully that the authors will add this part to the content.
In addition, although the authors have taken into account the contextual factors, they are also asked to think about the problems of assessment in different cultural structures. The authors also mention that the ELOM tool is the only standardized tool in South Africa and that its cross-cultural fairness has been proven, and that it can be used to measure developmental domains, which shows that the authors have attached great importance to this topic. Moreover, it is hopeful that the authors add specific practices to overcome the problems in different cultural structures in the paper.
Author Response
Please see attachment - includes response to all reviewers

Reviewer 2 Report
the methodology is unclear to me; authors should describe the research methodology more clearly and in a systematic way (both in the abstract and the main article).
Findings can be illustrated in the form of interesting diagram/figures.
Author Response

(The authors gave the same response as above.)

Reviewer 3 Report
Title:
Using data tools and systems to drive change in early childhood education for disadvantaged children: A South African case study.
The reviewer’s comments
The subject matter of this theme is good and well worth pursuing. However, the reviewer would like to see some revisions made to your manuscript.
1.The abstract will be revised to include following: Background, Context, Method, Data collection and conclusion. The author has written a lengthy sentence, therefore please modify the abstract.
2. Recommend the authors to think what new insight is your study offering to readers?
3. Introduction: Does the first paragraph serve as a good introduction?
Introduction needs to be updated to include the following: Problem of the study, Research Gap, Objective/Motivation, Hypothesis and research questions.
4. In the section of the discussion, I suggest the author should provide more theoritical literatures to dialogue with the results. Additional relevant studies should be included in order to enhance discussion.
5. Please strengthen the conclusion and implications. Good finding suggestions for future practitioners and researchers.
6. This study is interesting and innovative. Review again after major revision.

Title:
Using data tools and systems to drive change in early childhood education for disadvantaged children: A South African case study.
The reviewer’s comments
The subject matter of this theme is good and well worth pursuing. However, the reviewer would like to see some revisions made to your manuscript.
1.The abstract will be revised to include following: Background, Context, Method, Data collection and conclusion. The author has written a lengthy sentence, therefore please modify the abstract.
2. Recommend the authors to think what new insight is your study offering to readers?
3. Introduction: Does the first paragraph serve as a good introduction?
Introduction needs to be updated to include the following: Problem of the study, Research Gap, Objective/Motivation, Hypothesis and research questions.
4. In the section of the discussion, I suggest the author should provide more theoritical literatures to dialogue with the results. Additional relevant studies should be included in order to enhance discussion.
5. Please strengthen the conclusion and implications. Good finding suggestions for future practitioners and researchers.
6. This study is interesting and innovative. Review again after major revision.
Author Response

(The authors gave the same response as above.)

Round 2
Reviewer 2 Report
no further comment from me
Reviewer 3 Report
Title:
Using Data Tools and Systems to Drive Change in Early Child- 2 hood Education for Disadvantaged Children in South Africa
The reviewer’s comments
Thanks to the author for the correction. Revisions or explanations are all made according to the suggestions of the reviewers. Accept in present form.

Title:
Using Data Tools and Systems to Drive Change in Early Child- 2 hood Education for Disadvantaged Children in South Africa
The reviewer’s comments
Thanks to the author for the correction. Revisions or explanations are all made according to the suggestions of the reviewers. Accept in present form.